# Charting Past, Present, and Future Research in the Semantic Web and Interoperability

Abderahman Rejeb [1], John G. Keogh [2,*], Wayne Martindale [3], Damion Dooley [4], Edward Smart [5], Steven Simske [6], Samuel Fosso Wamba [7], John G. Breslin [8], Kosala Yapa Bandara [8], Subhasis Thakur [8], Kelly Liu [9], Bridgette Crowley [9], Sowmya Desaraju [9], Angela Ospina [10] and Horia Bradau [2]

[1]  Department of Management and Law, Faculty of Economics, University of Rome Tor Vergata, Via Columbia, 2, 00133 Rome, Italy; abderahman.rejeb@students.uniroma2.eu
[2]  McGill Centre for the Convergence of Health and Economics (MCCHE), Montreal, QC H3C 3J7, Canada; horia.bradau@nutriscope.ca
[3]  National Centre for Food Manufacturing, Centre of Excellence Technology Park, University of Lincoln, Holbeach, Spalding PE12 7LD, UK; wmartindale@lincoln.ac.uk
[4]  Centre for Infectious Disease Genomics and One Health, Simon Fraser University, Burnaby, BC V5A 1S6, Canada; damion_dooley@sfu.ca
[5]  Institute of Industrial Research, University of Portsmouth, Portsmouth PO1 2DY, UK; edward.smart@port.ac.uk
[6]  Systems Engineering Department, Colorado State University, Fort Collins, CO 80523, USA; steve.simske@colostate.edu
[7]  Department of Information, Operations and Management Sciences, Toulouse Business School, 31068 Toulouse, France; s.fosso-wamba@tbs-education.fr
[8]  Data Science Institute, National University of Ireland Galway, H91 TK33 Galway, Ireland; john.breslin@nuigalway.ie (J.G.B.); kosalayb@gmail.com (K.Y.B.); subhasis.thakur@insight-centre.org (S.T.)
[9]  McGill MCCHE Summer Research Intern Program, Montreal, QC H2Y 2E7, Canada; crkellyliu@gmail.com (K.L.); bridgette.crowley@icloud.com (B.C.); sowmyad96@gmail.com (S.D.)
[10] Institut D'études Européennes (IEE), Université Libre de Bruxelles, 1050 Brussels, Belgium; ajospinae@gmail.com
*  Correspondence: john@shantalla.org

**Abstract:** Huge advances in peer-to-peer systems and attempts to develop the semantic web have revealed a critical issue in information systems across multiple domains: the absence of semantic interoperability. Today, businesses operating in a digital environment require increased supply-chain automation, interoperability, and data governance. While research on the semantic web and interoperability has recently received much attention, a dearth of studies investigates the relationship between these two concepts in depth. To address this knowledge gap, the objective of this study is to conduct a review and bibliometric analysis of 3511 Scopus-registered papers on the semantic web and interoperability published over the past two decades. In addition, the publications were analyzed using a variety of bibliometric indicators, such as publication year, journal, authors, countries, and institutions. Keyword co-occurrence and co-citation networks were utilized to identify the primary research hotspots and group the relevant literature. The findings of the review and bibliometric analysis indicate the dominance of conference papers as a means of disseminating knowledge and the substantial contribution of developed nations to the semantic web field. In addition, the keyword co-occurrence network analysis reveals a significant emphasis on semantic web languages, sensors and computing, graphs and models, and linking and integration techniques. Based on the co-citation clustering, the Internet of Things, semantic web services, ontology mapping, building information modeling, bioinformatics, education and e-learning, and semantic web languages were identified as the primary themes contributing to the flow of knowledge and the growth of the semantic web and interoperability field. Overall, this review substantially contributes to the literature and increases scholars' and practitioners' awareness of the current knowledge composition and future research directions of the semantic web field.

**Keywords:** semantic web; interoperability; ontology; internet of things; semantic web services; bioinformatics; building information modeling; bibliometric

## 1. Introduction

Globalization, collaboration, and co-operation have significantly altered the software industry and led to the sharing of knowledge in vast, open settings [1]. A single human user cannot develop competencies and knowledge unless he or she collaborates with other human users and businesses. Consequently, the difficulty of knowledge representation is elevated to a central position [2,3]. Any user, including software agents, robots, smart devices, and humans, should use, generate, and exchange knowledge in expansive and heterogeneous environments. A common language and structure for expressing knowledge are utilized to address this issue. The semantic web as an infrastructure of computer-interpretable structured information [4–6] is a new technological development for describing and storing knowledge on the web [7,8].

The semantic web arose from the desire to provide query and comparison capabilities for all World Wide Web data standardized by the World Wide Web Consortium (W3C), allowing it to link URLs to resources and classify specific knowledge domains. Semantic web technology derives meaning from a structured hierarchy of data classification, like an encyclopedia's collective volumes, and is used to develop knowledge domains. Classification techniques for unstructured data are the initial step toward defining quantitative and qualitative relationships between data so that they can be queried and analyzed using semantic web technologies. According to [9], the semantic web is an extension of the existing web, in which knowledge is assigned a well-defined and unambiguous meaning through ontologies [10,11]. Ontologies are essential for achieving interoperability, as they provide structured vocabularies with a formal specification of shared concepts [12–14]. By providing a shared understanding of a particular topic of interest, ontologies aid in overcoming the problems caused by semantic heterogeneity. However, the most significant barrier to data integration and interoperability continues to be matching ontologies. The research on the semantic web demonstrates how ontologies can be utilized to resolve interoperability issues at the application level. Consequently, ontologies have been used during the discovery process to express the capabilities of the services. Similarly, ontologies improve user communication by defining the semantics of the symbolic representations used in the communication process.

Numerous scholars have examined the applications of semantic web technologies in various fields. For instance, in education, ref. [15] argues that the semantic web's reliance on ontologies that provide machine-interpretable information can facilitate the realization of "anybody, anytime, anywhere" learning. Similarly, ref. [15] proposes developing a framework for personalized e-learning based on domain ontology and aggregate user profiles. The entire procedure is divided into two stages: offline, which includes data preparation, ontology development, and usage mining; and online, which includes the production of recommendations. The authors of [16] describe 18 ontologies based on different states of human behavior such as emotion, needs, and mood. According to the authors, ontology is the actual representation of knowledge in a format that a computer can easily interpret. The authors of [17] describe the evolution of linked data on the web and its applications. The authors illustrate several web-based data publication methods. Recognizing the interoperability issue between systems and applications, ref. [18] explains how to effectively leverage semantic web technologies (SWTs) to address security and interoperability issues. In addition, some academics have conducted research on semantic web applications in knowledge discovery and data mining. These include [19,20], which summarize research on semantic web mining and demonstrate the efficacy of semantic web in enhancing web mining results. Similarly, ref. [21] examines research on the semantic-based solution for web mining and argues that the semantic web can simplify the extraction of pertinent documents.

Academics have also investigated semantic web technologies in the Internet of Things (IoT) and artificial intelligence (AI) systems. To illustrate, ref. [22] examines recent devel-

opments in the construction of knowledge bases and discusses knowledge-based fault diagnosis for industrial IoT systems. The authors of [23] discuss semantic interoperability in the web of things and recommend ontology learning and alignment techniques. Similarly, ref. [24] proposes a use case diagram for the alignment of IoT entities and discusses ontology's role in these entities' abstraction and semantic integration. The authors of [25] address the issues of limited depth and expressiveness in current ontologies by developing an enhanced reference generalized ontological model based on a reference architecture model for Industry 4.0.

Using an AI-related artificial neural network (ANN) model, ref. [26] proposes a novel algorithm for ontology matching. The authors of [27] explain the benefits of combining ANNs and semantic web technologies. In addition, ref. [28] emphasizes the capacity of the semantic web to accommodate computational intelligence, such as fuzzy logic, ANN, and evolutionary computation. According to the authors, supervised and unsupervised ANNs can be used for ontology alignment and learning. In the semantic web, fuzzy logic can also improve query results. Semantic web technologies can also be used to address cloud computing problems. In this regard, ref. [29] provides an overview of semantic information processing at the web scale for cloud computing. The authors of [30] also analyze the areas where semantic models can benefit cloud computing. Their findings indicate that semantic models have three application areas in cloud computing: functional and nonfunctional definitions, data modeling, and service prescription enhancement. Finally, ontologies for cloud computing were reviewed by [31]. The authors summarized research into four categories: cloud services discovery and selection, cloud interoperability, cloud resources and service prescription, and cloud security.

As research fields become increasingly complex and mature, scholars should periodically evaluate the knowledge built and accumulated to identify new contributions, capture trends and research traditions, comprehend which themes are discussed and the theories applied, and propose future research directions [32]. Knowledge domain visualization [33] aims to reveal hidden patterns in the formation and structure of scholarly knowledge through graphical illustrations [34]. This is accomplished by mapping the entire domain of scientific knowledge pertaining to the semantic web and interoperability. Even though bibliometric techniques have been utilized in a variety of fields to examine the development of knowledge [35–39], no prior research has examined the intellectual structure of research at the intersection of the semantic web and interoperability. We believe the current study makes significant contributions to the existing body of knowledge by bridging this gap. First, we argue that this is the first study to use bibliometric techniques to review semantic web and interoperability research by analyzing information about authors, studies, and keywords. By analyzing bibliometric networks, dynamics, and eminent scholars in this knowledge field, we contribute to the discussion on the semantic web's future directions, opportunities, and emerging challenges.

Second, we argue that the enormous amounts of data collected instantaneously in global business activities have the potential to be transformative in improving the human condition and provide solutions to many global and local issues concerned with a fair and equitable distribution of resources, wealth generation, and opportunity. However, the data need to be standardized through standards such as the W3C [40]. This is because data utilization requires sources to be linked or interoperable to be accessible by end-users. This standardization and interoperability can be addressed by developing the knowledge graphs used by organizations such as Google, Uber, Amazon, Facebook, and Netflix. A novel approach is to create a Decentralized Knowledge Graph (DKG), which has been made possible by combining graph technology and blockchain technology. Blockchain provides a consensus layer that verifies the authenticity of the data entering the DKG. It is gated by an identifier of source (i.e., data provenance) and conforms to W3C standards. Smaller digital technology firms such as Slovenia-based OriginTrail have developed an open-source ecosystem comprising tools and protocols that connect blockchain to blockchain and blockchain to legacy. While OriginTrail has numerous case studies [41,42] they have

migrated towards implementing a DKG with full compliance to W3C and GS1 standards in supply chains. A significant development is the transition from stove-piped data held in organizational siloes to interoperable Web3-based ecosystems where multiple datasets can be discovered, queried, and integrated. Thus, this paper developed from the need to understand how the World Wide Web is currently using data and to define how the semantic web can use it. The research presented here will help define how the future development of a DKG can evolve using semantic web technology, and we posit that DKGs will be transformative for industrial and social systems. Finally, newcomers to this intriguing and relevant discipline will benefit from the study's self-contained nature.

More specifically, in this study, we aim to answer the following research questions:

1. How has semantic web and interoperability research evolved since its inception?
2. What countries are at the intersection and forefront of semantic web and interoperability research?
3. Which scholars are the most contributive to the field?
4. What kinds of collaborative relationships exist between countries and institutions in this field?
5. What is the present status of research, and what are the future directions in the semantic web and interoperability field?

The paper continues as follows. Section 2 presents the methodology used in the review. In Section 3, the descriptive results are analyzed, followed by the results from the network analysis in Section 4. In Section 5, we perform a content analysis based on co-citation clustering, and next, we discuss the review's findings, its limitations, and potential suggestions for future research. Finally, we briefly conclude the work.

## 2. Methodology

### 2.1. Research Methods

Arguably, the primary objective of a literature review is to identify, specify, map, and evaluate the current body of relevant literature in a systematic, objective, and replicable manner [43–46]. According to [47], conducting a structured literature review can ensure that a diverse range of publications and methodologies are covered and that they are analyzed thoroughly and in detail. The bibliometric approach was selected in this study for several reasons. First, the bibliometric approach is more reliable and scalable than other techniques for text analysis (e.g., content analysis) [48–50]. Second, bibliometric analysis enables researchers to better understand the subject by examining the relationships between selected papers, references, keywords, and co-citations, thereby providing a comprehensive picture of a particular research field [51–53]. Thirdly, bibliometric tools enable researchers to generate meaningful results and intuitively visualize the most significant cluster of research topics within a knowledge domain [54].

### 2.2. Data Collection

A preliminary search was conducted in the Scopus database on 1 June 2021, using the following search terms: interoperab* and "semantic web". In comparison to other academic databases (e.g., Web of Science), Scopus is well-known for its extensive coverage [55] and intuitive tools that enable researchers to efficiently obtain and compile references from a sample of documents [56]. Additionally, Scopus was chosen for its dependability and the volume of scholarly publications it indexes, including journals published by prestigious publishers such as Elsevier, Springer, Emerald Insight, IEEE; and, Taylor and Francis. To begin, the title, abstract, and keywords fields were filled with the search query. Three thousand six hundred and forty-one (3641) publications were returned as the initial result. The returned dataset was filtered to exclude publications from the incomplete year 2021 and include only English-language documents to ensure a consistent set of publications. Additionally, duplicated publications or publications that lacked bibliographic data (e.g., abstracts, keywords, or authors' names) were omitted. This search yielded 3511 publications

extracted from Scopus for additional analysis. These publications' references and citations were saved in CSV and txt formats suitable for direct text mining.

### 2.3. Initial Statistics for the Analysis

Regarding the initial statistics, we began by analyzing the evolution of research at the intersection of interoperability and the semantic web by plotting the selected set of publications according to their annual distribution. Following [36], we used several quantitative indicators to determine the impact and quality of research, including publications per author, citations per publication, and geographic origin of publication. To make data entry and processing more manageable, we analyzed the bibliometric data using the software package BibExcel. The strength of BibExcel comes from its compatibility with a wide variety of academic databases, including Scopus, and visualization tools, such as VOSviewer and Gephi.

Authors who receive many citations are considered influential in their respective research fields. To ascertain the authors' contribution to interoperability and the semantic web, we extracted information about the authors, their affiliations, and their frequency of appearance. The authors' affiliations were imported into BibExcel to identify the leading academic institutions and the countries in which they are located. To better understand the emphasis of the research on interoperability and the semantic web, we conducted a keyword analysis to determine the most frequently used term in the sample. It is critical to track citations and understand their trends to assess the impact and influence of research. The number of citations a publication receives indicates its degree of importance within the scholarly community, which is one possible indicator of the impact and influence of research [57]. We calculated the number of citations received by each publication from other publications in our sample, considering the frequency of local citations. The global citation count was also used to estimate the number of citations received by a single publication in Google Scholar. The latter is the most comprehensive academic database available, including Scopus, Web of Science, Springer, and many others. The difference between a publication's local and global citations reflects the level of interest generated within and across research fields.

### 2.4. Network Analysis

Following an analysis of descriptive statistics, the intellectual landscape of research at the intersection of interoperability and the semantic web was represented by networks of various entities, including countries, academic institutions, and authors. VOSviewer was used to create these networks from the extracted bibliographic sources. Additionally, we conducted an in-depth analysis of the trends and relationships between the publications in our sample. We performed network analysis on bibliometric data and visualized the network structure of interoperability and the semantic web using the visual tool Gephi. The strength and significance of a network connection between two publications can be explained by the co-occurrence of keywords in the same publication or the co-citation of the same publication [58].

The visualization of co-citation networks is an example of exploratory data analysis (EDA) that utilizes graph theory to examine the data structure [59]. A co-citation network comprises a set of nodes that represent publications and a set of edges that represent the co-occurrence of publications in the network's reference list [60]. Co-cited publications are those that appear in the reference lists of other publications. As a result, publications A and B are co-cited if publication C cites both publications A and B. It has been demonstrated that publications that are frequently cited together by other publications are more likely to be related and thus to contain similar content or subject matter [61]. This approach was used to map and categorize the literature on interoperability and the semantic web. Since a highly cited paper might not be highly prestigious, prestige is a critical impact indicator [62]. In this regard, PageRank was introduced to measure both the citability and

prestige of a paper [63]. In this study, we identified the leading ten papers from each cluster by their PageRank measure to ensure the analysis of high-quality papers in each cluster.

Gephi is used to generate the co-citation network from the BibExcel data. We used a procedure to generate the co-citation network cluster. For example, prior to using Gephi, bibliometric data were pre-processed, and a citation frequency threshold of 50 was established. Caution was exercised in this step, as a low threshold value may produce an excessive number of clusters, while a high threshold value may result in over-filtering. Additionally, to create a meaningful visualization, a simple and interpretable map was generated using the force atlas layout in Gephi, which is used to reposition the nodes in the network in a more simplistic and readable manner. Each node in the network represents a publication, and each connection between two nodes represents a co-citation relationship. We manually adjusted the hierarchies, node sizes, and other settings to generate the co-citation network (e.g., node color).

We conducted a keyword co-occurrence network analysis to better understand the interoperability and semantic web research. Like a co-citation network, a keyword co-occurrence network illustrates the co-occurrence of keywords and their relationships [64]. According to [65], analyzing this network enables researchers to identify research topics and document the transition of research frontiers within a particular field of study. If two keywords appear more frequently in the same publications, they have a stronger relationship. By creating the keyword co-occurrence network, we hoped to analyze the core content of the keywords used and better understand the research structure at the intersection of interoperability and the semantic web. VOSviewer was used to visualize the co-word network because it is highly recommended and compatible with the BiBExcel software package. The node's size corresponds to the frequency of each keyword, while the thickness of the edges indicates the frequency with which each pair of keywords appears in publications. This approach was used in the current study to describe and analyze the keyword co-occurrence network. Visualizing the associations between keywords made it possible to deduce the topics related to interoperability and the semantic web.

## 3. Analysis of Descriptive Results

Figure 1 demonstrates the publication trend since the year 2000. Between 2003 and 2006, the number of studies increased dramatically, and approximately 83% of publications were published after 2006. The year with the most publications was 2010, with 243. The first four papers published in 2000 concentrated on the roles of XML and RFD in the development of semantic webs [66,67]. A significant finding was that the first decade (2000–2010) of interoperability and semantic web research can be considered the infancy stage. The second decade (2011–2020) saw a steady level of interest in this subject, as indicated by the slight ups and downs in the number of publications. This period may be referred to as the stabilization and consolidation stage, as growth appears to have reached maturity and saturation during the second decade.

The distribution of publications by type is shown in Table 1. As can be seen, conference papers dominate the list, accounting for more than 60% of all publications selected. This is unsurprising, given that conference papers represent researchers' most recent accomplishments and indicate emerging research trends and the immediacy of personalized peer feedback [68]. As represented by journal articles, peer-reviewed literature is second on the list. Additionally, the sample includes 113 book chapters and 108 conference reviews.

Table 2 lists the top fifteen journals that have published articles on interoperability and the semantic web. These fifteen journals published 207 articles, accounting for nearly 21% of the 997 journal articles identified. The remaining 79% of the articles were dispersed across multiple outlets. The *International Journal of Semantic Web and Information Systems* and the *International Journal of Semantic Web* published 18 articles. The *International Journal of Metadata Semantics and Ontologies* and the *Journal of Biomedical Informatics* published 16 articles. By and large, these journals appear to be devoted to semantics and ontologies. Additionally, *IEEE Intelligent Systems*, *Biomedical Informatics*, *Automation in Construction*,

and *Expert Systems with Applications* have all published extensively on the subject. This group of journals appears to be devoted to computer science and the development of interoperable systems. As seen in the table, the core journals that play a critical role in knowledge dissemination are primarily concerned with semantics and information technology. Despite growing interest in these fields, it may take time for related fields such as social science, business, and management to pay attention. As a result, additional interdisciplinary research is required.

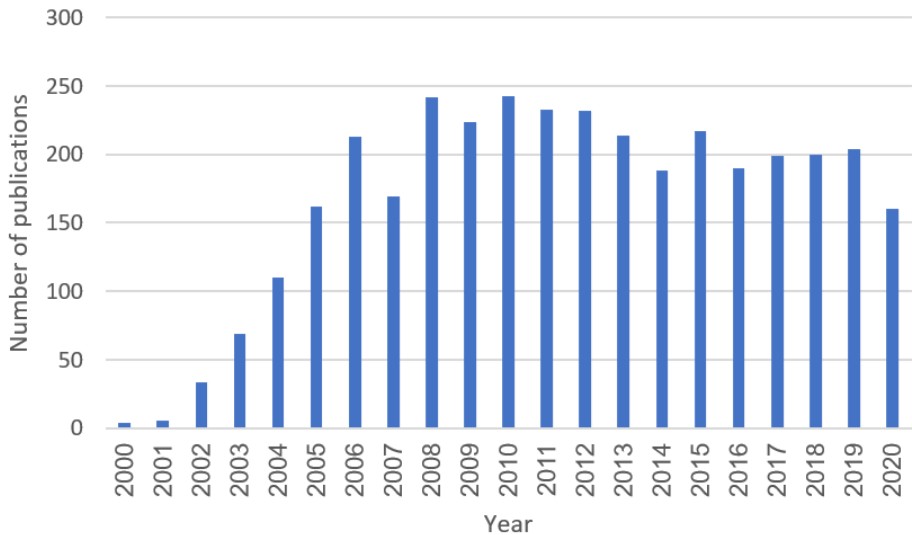

**Figure 1.** Year-wise distribution of publications using the interoperab* and "semantic web" search.

**Table 1.** Distribution of publications according to type.

| Document Type | Number |
|---|---|
| Conference Paper | 2273 |
| Article | 920 |
| Book Chapter | 113 |
| Conference Review | 108 |
| Review | 77 |
| Book | 12 |
| Editorial | 3 |
| Short Survey | 2 |
| Letter | 1 |
| Note | 1 |
| Undefined | 1 |

**Table 2.** Top 15 most relevant journals.

| Journal | Number of Publications |
|---|---|
| *International Journal on Semantic Web and Information Systems* | 18 |
| *Semantic Web* | 18 |
| *International Journal of Metadata Semantics and Ontologies* | 16 |
| *Journal of Biomedical Informatics* | 16 |
| *IEEE Intelligent Systems* | 15 |
| *Journal of Biomedical Semantics* | 15 |
| *Sensors (Switzerland)* | 13 |
| *Automation in Construction* | 12 |
| *Expert Systems with Applications* | 12 |
| *Journal of Universal Computer Science* | 12 |
| *Cataloging and Classification Quarterly* | 9 |
| *IEEE Internet Computing* | 9 |
| *Journal of Information Science* | 9 |
| *Journal of Web Semantics* | 9 |
| *IEEE Access* | 8 |
| *Journal of Theoretical and Applied Information Technology* | 8 |
| *Knowledge Based Systems* | 8 |

The top twenty academic institutions that contribute to interoperability and the semantic web are listed in Table 3. Authors from the United States and Germany contributed the most publications to the list, with 545 and 331 publications, respectively. The United Kingdom is third with 322 publications, while Italy is fourth with 303. China, India, and Brazil have all contributed significantly to the literature, publishing 250, 109, and 91 papers, respectively. The remaining papers in our dataset were contributed by scholars from Australia and several European and Asian countries.

**Table 3.** Top 20 most productive countries.

| Country | Number of Publications |
| --- | --- |
| United States | 545 |
| Germany | 331 |
| United Kingdom | 322 |
| Italy | 303 |
| Spain | 295 |
| France | 259 |
| China | 250 |
| Netherlands | 162 |
| Greece | 142 |
| Austria | 130 |
| Ireland | 121 |
| Canada | 115 |
| India | 109 |
| Australia | 107 |
| Brazil | 91 |
| Finland | 90 |
| Belgium | 84 |
| South Korea | 79 |
| Portugal | 77 |
| Switzerland | 63 |

## 4. Results from Network Analysis

The international collaboration network for interoperability and semantic web research is depicted in Figure 2. The radius of each node in this network represents the number of publications in that country, while the edge represents the level of collaboration between countries. We can see from the network that the most productive country, the United States, has a strong research collaboration with the United Kingdom, Italy, and Switzerland. The USA also collaborates with countries such as Slovenia, Bulgaria, Denmark, and Austria on a smaller scale. In short, while researchers in the fields of interoperability and the semantic web are distributed globally, Africa and Asia continue to be underrepresented. This could be due to several factors, including the isolation of the scholars in these continents from the international scientific community (e.g., conferences), the possibility that their publications are in non-indexed journals, or that their research activities are being conducted in their mother tongue. International collaboration and joint research efforts with scholars from the most productive countries are therefore required to improve the quality of research output and advance the field of interoperability and the semantic web.

The top twenty academic institutions that contribute to the literature are listed in Table 4. Scholars published the highest number of articles at the National University of Ireland Galway. Compared to the top contributors in Table 5, the Open University, the CNRS Centre National de la Recherche Scientifique, and Stanford University are all represented by the most prolific authors: Domingue J., Gyrard A., and Dumontier M., respectively.

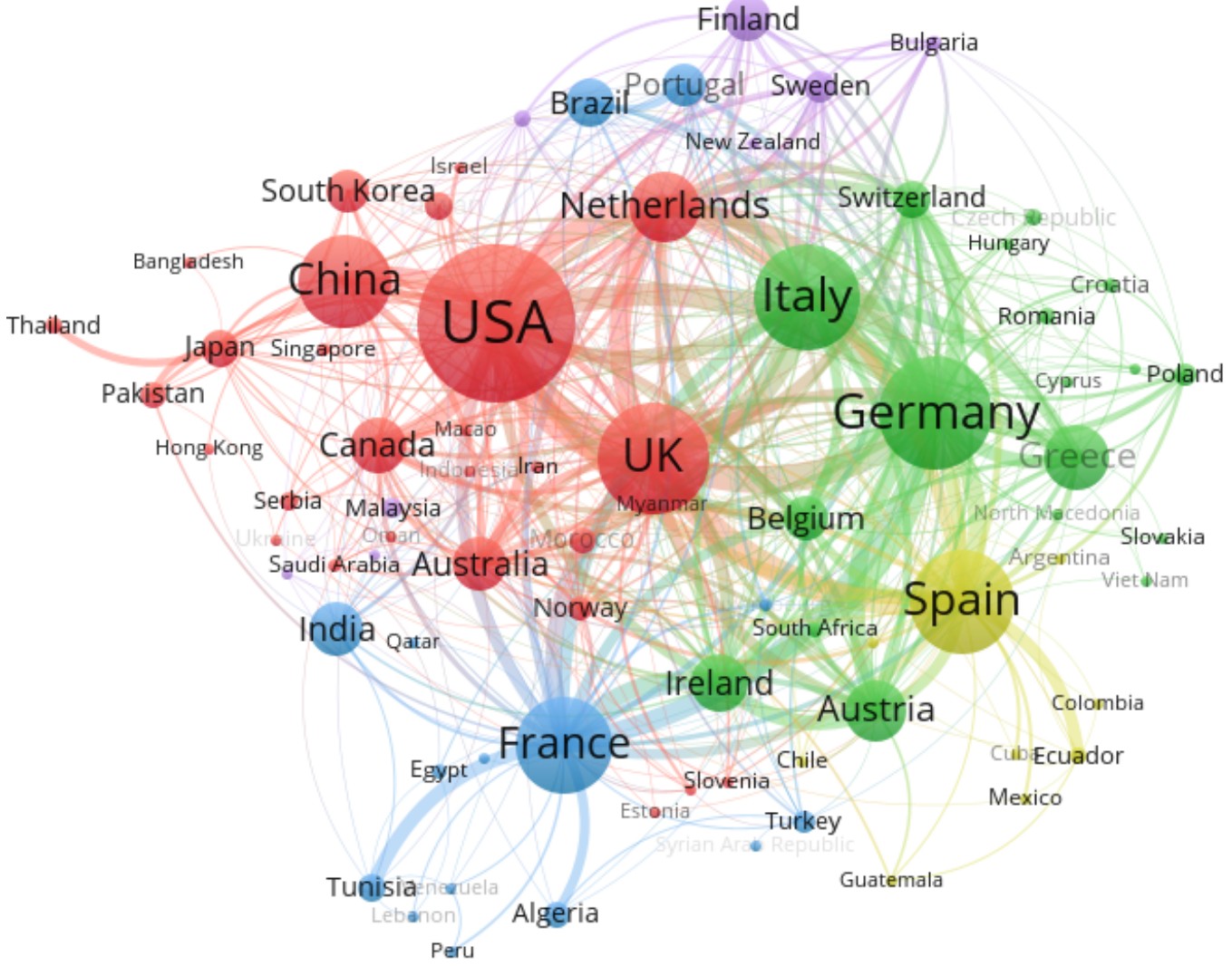

**Figure 2.** Network of international collaboration for interoperability and semantic web research.

The network of institutional collaboration in the field of interoperability and the semantic web is depicted in Figure 3. The node size is proportional to the number of publications produced by each institution, whereas the thickness of the edges indicates the level of research collaboration. The network demonstrates that institutions are assigned to several research partitions based on the color scheme used. The largest cluster is red, with 32 institutions spread across North America, Europe, and Asia. Universidad Politécnica de Madrid is the most productive institution in this group, followed by Vrije Universiteit Amsterdam and Universiteit Gent. The green cluster is the second largest. The most productive institution is the National University of Ireland Galway, followed by the Digital Enterprise Research Institute and the University of Southampton. (The Digital Enterprise Research Institute was a research institute at the National University of Ireland Galway, from 2003 to 2013. Between 2003 and November 2007, the Digital Enterprise Research Institute brand was also shared by the University of Innsbruck.) The blue cluster is the third largest, consisting of the most productive institution, the Consiglio Nazionale delle Ricerche, and the Alma Mater Studiorum Universita di Bologna. While the network demonstrates strong collaboration among UK universities, we observe only sporadic research collaboration with other countries. To summarize, this type of research collaboration among institutions working in the field of interoperability and the semantic web is indicative of the knowledge domain's fragmented nature and a dearth of research collaboration among individual scholars.

**Table 4.** Top 20 most productive institutions.

| Institution | Number of Publications |
|---|---|
| National University of Ireland Galway | 69 |
| Universidad Politécnica de Madrid | 63 |
| Digital Enterprise Research Institute | 46 |
| CNRS Centre National de la Recherche Scientifique | 41 |
| Consiglio Nazionale delle Ricerche | 38 |
| Alma Mater Studiorum Università di Bologna | 36 |
| The Open University | 35 |
| Vrije Universiteit Amsterdam | 34 |
| University of Southampton | 34 |
| Universiteit Gent | 33 |
| Insight Centre for Data Analytics | 31 |
| The University of Manchester | 30 |
| Stanford University | 28 |
| Wuhan University | 27 |
| Karlsruhe Institute of Technology | 27 |
| Technische Universitat Wien | 26 |
| Technische Universiteit Eindhoven | 26 |
| University of Innsbruck | 25 |
| Aalto University | 24 |
| Universidad de Murcia | 23 |
| Universidad Carlos III de Madrid | 23 |
| Aristotle University of Thessaloniki | 23 |

**Figure 3.** Network of institutional collaboration.

The authors listed in Table 5 are the most prolific contributors to the pertinent literature. As can be seen, Domingue J. and Gyrard A. co-authored 19 papers, placing them among the top authors with the most publications. It is worth noting that Domingue J. and Dietze S. and Gyrard A. and Serrano M. co-authored many of these articles. Almost all of these researchers have experience with semantic web technologies. For instance, Domingue J. frequently publishes his work in books and at conferences, focusing on the semantic web's suite of technologies, including semantic web services. The proliferation of his papers may be attributed to the potential benefits of semantic web services and the critical nature of interoperability in automating the use of information and ensuring its reliable exchange between service providers and customers. Other authors, on the other hand, tend to construct an ontology for the Internet of Things (IoT). Gyrard A., for example, conducts a thorough analysis of current IoT-related ontologies, proposes solutions for semantic interoperability across multiple testbeds, and designs interoperable ontology-based IoT applications. The diverse methodologies and interests of the most productive authors, in general, reflect the interdisciplinary and rich nature of interoperability and semantic web research.

**Table 5.** Top 20 most productive authors.

| Author | Number of Publications |
| --- | --- |
| Domingue, J. | 19 |
| Gyrard, A. | 19 |
| Dumontier, M. | 17 |
| Sheth, A. | 17 |
| Dietze, S. | 16 |
| Bassiliades, N. | 15 |
| Decker, S. | 15 |
| Serrano, M. | 15 |
| Fernández-Breis, J.T. | 14 |
| Mannens, E. | 14 |
| Verborgh, R. | 14 |
| Christodoulakis, S. | 13 |
| Gómez-Pérez, A. | 13 |
| Piedra, N. | 13 |
| Thuraisingham, B. | 13 |
| Breslin, J.G. | 12 |
| Loia, V. | 12 |
| Terziyan, V. | 12 |
| Wilkinson, M.D. | 12 |
| Finin, T. | 11 |
| García-Castro, R. | 11 |
| O'Sullivan, D. | 11 |

The author collaboration network in the field of interoperability and the semantic web is depicted in Figure 4. The size of each node in this network corresponds to the total number of publications by each author. In contrast, the thickness of the edges is proportional to the number of co-authored publications. A highly dispersed network is indicative of the authors' lack of close collaboration in the field under study. The authors are clustered into 60 research communities via the network. These authors are either part of a few scattered co-authorship teams or are entirely disconnected from the other authors, implying that co-authorship is uncommon in this research field. As a result, the most productive authors are more likely to be associated with dense communities with weak co-authorship associations outside their scientific neighborhoods. By and large, the structure of the co-authorship network in the field of interoperability and the semantic web is comparable to that of other networks in other domains of knowledge [69,70].

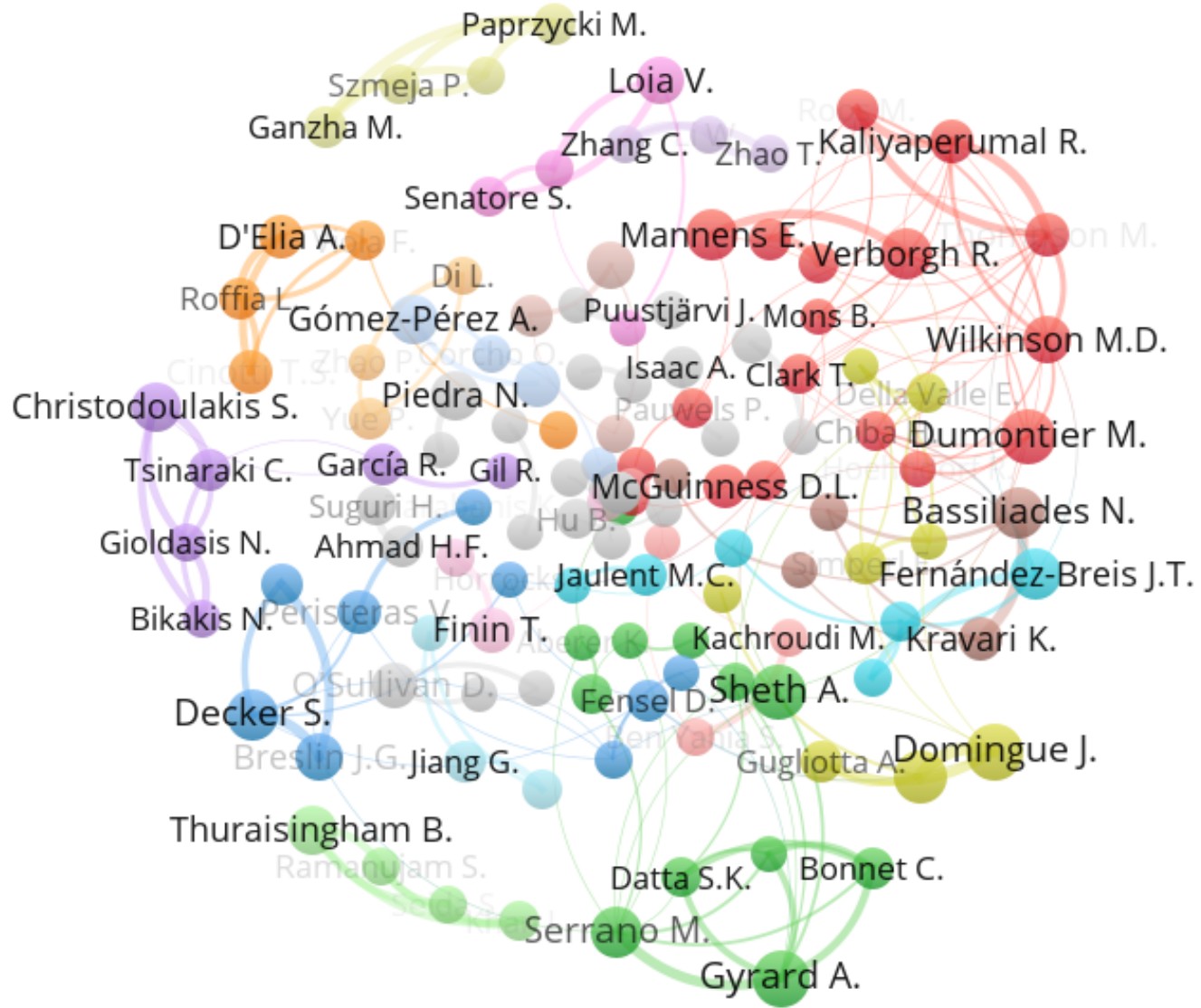

**Figure 4.** Co-authorship network.

The following table (Table 6) summarizes the top twenty most frequently used keywords. As can be seen, "SW" (semantic web) is the most frequently occurring keyword, owing to its inclusion in the search query. The keyword "ontology" appears as the second most frequently used. As a result of this finding, designers, users, and domain experts must agree on shared and reusable conceptualizations and knowledge to comprehend the actual discourse in a particular domain. Additionally, the terms "interoperability" and "SI" (semantics interoperability) appear on the list, indicating that interoperability is a critical component of increasing the usability of distributed information systems, enabling structured search and data sharing and laying the groundwork for higher-level (web3) services and processing [71]. Similarly, semantic interoperability can be used to ensure the interoperability of IoT devices from various suppliers to reduce costs associated with data analysis and facilitate rapid decision making [72].

**Table 6.** Top 20 most frequent keywords.

| Keyword | Frequency |
|---|---|
| SW (semantic web) | 1004 |
| Ontology | 784 |
| Interoperability | 443 |
| SI (semantics interoperability) | 239 |
| RDF (resource description framework) | 200 |
| Linked data | 195 |
| IoT (Internet of Things) | 184 |
| WS (web services) | 144 |
| OWL (ontology web language) | 142 |
| SWS (semantic web services) | 136 |
| Semantics | 100 |
| Metadata | 98 |
| SOA (service-oriented architecture) | 73 |
| LOD (linked open data) | 66 |
| Data integration | 64 |
| SPARQL (simple protocol and RDF query language) | 60 |
| Semantic web technology | 59 |
| XML (extensive markup language) | 57 |
| MAS (multi-agent systems) | 54 |
| KM (knowledge management) | 53 |

According to Gephi's citation analysis, the selected publications cited one another. According to both local and global citations, the top ten most cited publications are listed in Table 7. Among all publications, ref. [73] received the most citations. Two researchers, Sheth and Decker, listed in Table 5 as top contributors, also have publications in the Table 7 list. This finding is surprising, because the most prolific authors' influence is still limited compared to the authors listed in Table 7. As a result, no significant relationship exists between the number of publications and the total number of citations received. Surprisingly, two publications have extremely high global citation counts: [66] has 1189 citations and [74] has 703 citations, even though their local citation counts are relatively low. These two publications discuss the role of ontologies in the semantic web's architecture and their use in organizing registries and classifying all web services. These results demonstrate that these studies laid the groundwork for subsequent publications in the fields of interoperability and semantics. A closer examination of Table 7 reveals that Roman, Grosof, and Decker all have ontology, semantic technologies, knowledge management, and artificial intelligence backgrounds. This may account for their high global citation count.

**Table 7.** Top ten most cited articles with local and global citations.

| Publication | Local Citations | Global Citations |
|---|---|---|
| [73] | 670 | 1493 |
| [75] | 603 | 1344 |
| [76] | 414 | 772 |
| [66] | 385 | 1189 |
| [77] | 376 | 695 |
| [78] | 367 | 690 |
| [79] | 350 | 722 |
| [80] | 334 | 615 |
| [81] | 321 | 513 |
| [74] | 302 | 703 |

Keyword co-occurrences constitute a relational bibliometric indicator that depicts academic knowledge. By analyzing keyword co-occurrence networks, scholars can determine the clusters that reflect a comprehensive view of diverse research foci in a specific research domain. We started by retrieving all keywords from the selected publications to obtain the

network. Then, we preprocessed and refined the keywords to ensure homogeneity, consistency, and accuracy. For example, full-length keywords such as semantic web, semantic interoperability, and the Internet of Things were abbreviated. In the visualization map, two keywords in close proximity are assumed to share a similar research topic or direction.

To generate the keyword co-occurrence network, the data were loaded into VOSviewer, and the density-based spatial clustering using the full-counting method was employed [82,83]. Additionally, the threshold of keyword occurrences was set at four. As a result, six clusters with different colors were obtained, as depicted in Figure 5. Each node in the figure corresponds to a keyword, and the node size reflects the co-occurrence frequency of the keyword. The density determines the distance between two nodes, and the higher this density, the shorter the distance between two keywords. The details regarding each cluster are presented in Table 8.

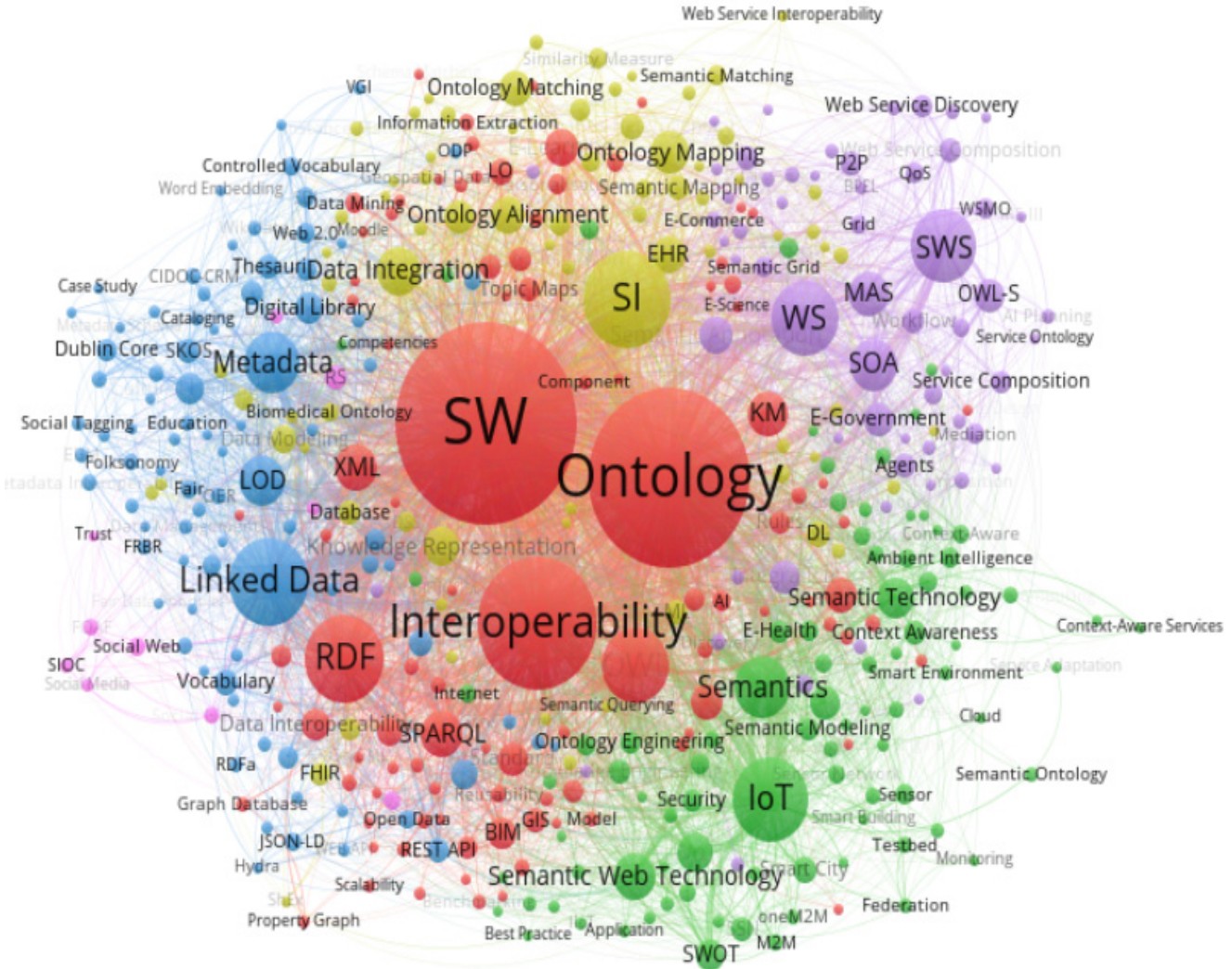

**Figure 5.** Keyword co-occurrence network.

From Figure 5, it is evident that the largest cluster is the red one. This cluster focuses on the fundamental building blocks of the semantic web, including ontologies and languages. The languages and technologies used for the construction of the semantic web provide valuable instruments for representing the semantics of profile data [84]. Ontologies and metadata languages play a significant role in integrating statical data on the web and annotating services [85]. In the semantic web context, several approaches are devoted to providing processable semantics expressed in meta-models, such as RDF, OWL, and OIL. The second cluster concerns sensors and computing. In this cluster, studies highlight

that sensor networks in the Internet of Things will be used by distinct entities and result in great heterogeneity [86]. The third cluster focuses on graphs and models. Frequent terms in this cluster include "Linked Data", "Metadata", "LOD" (Linked Open Data), "Big Data Knowledge", "Vocabulary", and "Thesauri". The application of linked data can be a pertinent solution to solve interoperability issues and ensure data consistency across the data sources [87]. The fourth cluster is labeled "Linking/Integration Methods", which indicates that researchers devoted importance to the tools used to facilitate data integration and information transformation [88–90]. These include ontology mapping, alignment, and matching. The next cluster focuses on semantic web services, representing a research field that attempts to apply semantic technologies to the description and use of web services. This will increase the machine understanding of web services and more effective communication. The high frequency of "E-Government" in this cluster indicates that specialized applications of the semantic web, such as XML schema and web service interfaces, can be leveraged to improve the interoperability of e-government information systems and simplify cross-organizational communication in a cross-border step. Finally, the last cluster deals with the potential of the semantic web to make social websites more interoperable. According to [91], the implementation of semantic web frameworks such as SIOC (Semantically Interlinked Online Communities) and FOAF (Friend-of-a-Friend) to the social web can lead to a social semantic web and the creation of a network of interlinked and semantically rich knowledge.

**Table 8.** The top 10 most frequent keywords in each cluster.

| Language | Sensors and Computing (Hardware) | Graphs and Models | Linking/Integration Methods | (Web) Services | Social Web |
|---|---|---|---|---|---|
| SW | IoT | Linked Data | SI | WS | RS |
| Ontology | Semantics | Metadata | Data Integration | SWS | Privacy |
| Interoperability | Semantic Web Technology | LOD | Ontology Mapping | SOA | Social Web |
| RDF | Semantic Technology | Cultural Heritage | Knowledge Representation | MAS | FOAF |
| OWL | WoT | Digital Library | Ontology Alignment | Semantic Annotation | Social Network |
| SPARQL | Cloud Computing | Big Data | EHR | E-Government | Social Networking |
| XML | SWOT | Knowledge Graph | Ontology Matching | OWL-S | SIOC |
| KM | Smart City | Vocabulary | IR | Integration | WEB APPS |
| E-Learning | Context Awareness | Thesauri | Data Modeling | Agents | Social Media |
| Reasoning | Knowledge Engineering | REST API | DL | Intelligent Agent | Trust |
| SWRL | SSN | SKOS | | P2P | |
| | | Web of Data | | Web Service Composition Workflow | |

## 5. Co-Citation Clustering Analysis

Following previous recommendations [92], we chose a co-citation frequency threshold of two and a citation count threshold of two. The visualization was cleared of all isolated nodes. This approach resulted in a co-citation network of 142 publications, a reduction of 3369 from the original 3511 publications. The network's nodes can be clustered into distinct communities, with a greater density of edges between nodes belonging to the same community than between nodes belonging to different communities [93]. Each community in the network represents a collection of closely related publications pertaining to interoperability and semantic web research, with only a tenuous connection to publications clustered in other communities (see Figure 6). Clustering publications enable an analysis of the network's structure, revealing topics, interrelationships, and collaboration patterns.

We used Gephi's default modularity tool based on the Louvain algorithm to create the co-citation network. This model is iterative, and the algorithm can determine the optimal number of communities that maximizes the modularity index [94]. A community's modularity index is a numeric value between −1 and 1 that indicates the density of connections within the community compared to connections between communities [95].

When this algorithm is used, eight clusters are generated. Each cluster has a different number of publications, ranging from four in cluster eight to thirty-one in cluster one, the latter being the largest. The modularity index in Figure 6 is equal to 639, indicating the existence of significant interrelationships between the eight clusters. Since closely related publications share similar characteristics, a cluster with a strong co-citation association reflects similar subject areas [96]. A closer examination of each publication in the same cluster reveals the cluster's primary research focus. Due to the large number of publications in each cluster, we decided to conduct a content analysis of the top ten publications in each cluster. We could identify and label clusters' research hotspots based on these publications. Table 8 summarizes the leading publications from each cluster. The research topics for each of the eight clusters are listed in Table 9.

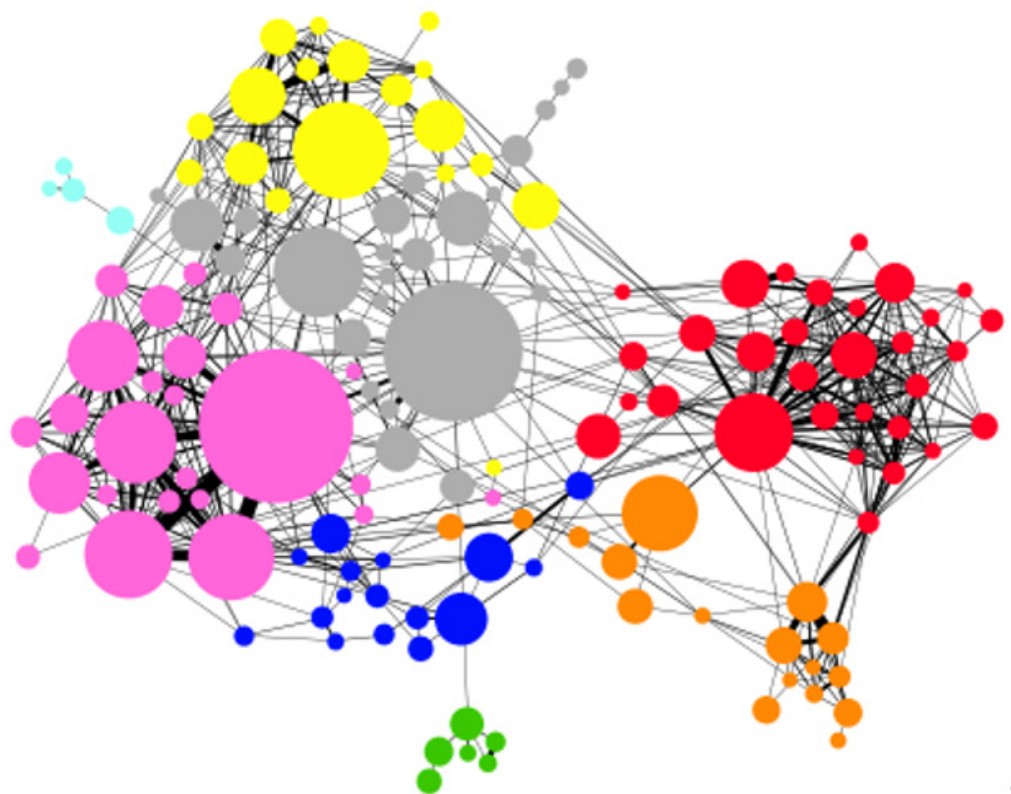

**Figure 6.** Co-citation network.

**Table 9.** Research focuses and number of publications in co-citation clusters.

| Cluster | Number of Publications | Research Focus |
|---|---|---|
| 1 | 31 | Internet of Things |
| 2 | 26 | Conceptualization of semantic web |
| 3 | 24 | Semantic web services |
| 4 | 18 | Ontology mapping |
| 5 | 17 | Building information modeling |
| 6 | 16 | Bioinformatics |
| 7 | 6 | Education and e-learning |
| 8 | 4 | Semantic web languages |

## 5.1. Internet of Things (IoT)

According to the categorization of research topics summarized in Table 9, research on interoperability and the semantic web places a premium on the importance of the Internet of Things (IoT) in developing the future internet. The IoT requires collaborative efforts from various stakeholders, including the telecommunications industry, device manufacturers,

the semantic web, and the computer science and engineering sectors [80]. The authors of [80] argue that the heterogeneous nature of the IoT presents several interoperability challenges, as global IoT implementation is not feasible. With the recent advancement of the IoT, the volume, velocity, variety, and volatility of data generated by IoT devices pose several challenges to established information systems. As a result, it is critical to integrate IoT technologies semantically and interoperably. Numerous studies have proposed various solutions for ensuring the semantic interoperability of Internet of Things (IoT) services. The authors of [97], for example, propose a first-of-its-kind open-source IoT platform that includes various visual tools for developing and implementing IoT applications with little or no programming. The authors of [86] propose a new semantic-level interoperability architecture for the Internet of Things and ubiquitous computing. The architecture's fundamental premise is to partition the global IoT into smaller, more manageable smart spaces. Similarly, the authors of [98] develop a lightweight semantic model for the IoT, adhering to ten rules for designing a good and scalable semantic model and optimizing the IoT in terms of memory requirements, computational time, processing time, and latency. The authors of [99] adopt partial differential fuzzy unsupervised models for semantic association decision analysis that links sensor data with associated web data. The authors of [100] present an architecture that links the IoT and semantic modeling within an Industry 4.0 context. It proposes an enhanced reference generalized ontological model for Industry 4.0 activities such as improved asset monitoring, process optimizations, and production enhancement. The authors of [101] present a solution for semantic interoperability among heterogeneous testbeds. As a result, the cluster's primary insight is the potential for future research into the formal methodologies required to develop scalable and interoperable IoT architectures and standard data formats. The proposed designs for the Internet of Things could be enhanced to achieve syntactic interoperability and security [72]. The literature is strikingly silent on how to account for the dynamic nature of physical environments and the difficulties associated with IoT resources when developing semantic tools and techniques [80].

*5.2. Conceptualization of the Semantic Web*

From clusters two to six, research focuses on the conceptualization, foundations, components, methodologies, and solutions necessary to improve semantics and interoperability. They are classified as semantic web conceptualization (cluster two), semantic web services (cluster three), ontologies (cluster four), and building information modeling (clusters five and six). The final two clusters, seven and eight, have fewer than ten publications, indicating that academia pays less attention to education, e-learning, and knowledge representation languages. Related studies on these subjects are sparse, necessitating the production of comprehensive works.

Cluster two contains publications on the semantic web's conceptualization and thus is titled "conceptualization of the semantic web". The authors of [66] argue that establishing higher levels of interoperability will make the semantic web feasible. The semantic web's vision is to create a global web in which data are defined by rich semantics and applications. According to [102], the semantic web is based on the concept of a shared and minimal language that enables the analysis and processing of massive amounts of existing data. The semantic web enables the capture and utilization of term meanings. As a way to augment the abilities of machines to communicate the meaning of information, the semantic web ushers in a more generally applicable approach to encoding any assertion of interest using the combination of RDF and extensible markup language (XML) to formulate the main elements of ontologies on the semantic web. Although the conceptualization of the semantic web has been reported numerous times in the existing literature, the ability of the semantic web to conceptually organize information and be coupled with advanced technology and smart spaces to create a digital service-oriented context for various fields has not been reported.

### 5.3. Semantic Web Services

The third cluster concentrates on bringing semantics to web services [77]. The fusion of semantic web and web services technologies aims to facilitate the automation of service usage by describing and annotating the different aspects of web services through explicit and machine-understandable semantics, thereby allowing the automatic location, integration, and use of web services [103]. The need for more knowledge-based and agent-based business communication and e-services has resulted in increased research on semantic web services. For instance, ref. [73] notes that web services could be enriched with machine-processable semantics to develop dynamic, scalable, and cost-effective marketplaces and e-commerce solutions. Semantic web services are expected to offer sophisticated web service development tools that enable automated simulation and verification of web service attributes and consistency-checking and debugging features [104]. The semantic web is helpful in overcoming the interoperability issues in diverse and complex web services. For example, semantic web services are leveraged to establish a social networking website for AEC projects and develop energy analysis applications [105]. The introduction of semantic descriptions and specifications has revolutionized web service technologies, providing extensive explanations of service contents, the automation of service selection, the exchange and translation of message content, the self-description of service functionalities, and recovery from failure. While semantic web services are crucial for bringing automatic service discovery, there is still a need to examine the role of this paradigm in ensuring more interoperable and semantically transparent architecture for some fields such as bioinformatics [106]. Biological data are challenging to integrate due to their complexity and inter-related nature; consequently, attempts to use semantics in bioinformatics web services may reduce the incompatibility of data standards, ensure data integration, and increase granularity for both data and services. There are few studies on the potential of semantic enrichment and semantic web services to integrate heterogeneous information from different data sources and address several issues in terms of interoperability, topology relationships, and extensions to standard schemas [107]. Additional studies considering the specificities of different application domains and user feedback during service composition are also required to ensure more automatic and user-friendly semantic web service composition for expert domains [108].

### 5.4. Ontology Mapping

Ontology standards such as RDF and DAML are essential to offer semantic context-based applications to users. In light of this, the fourth cluster represents the research perspective focused on ontology mapping. For example, ref. [76,88] study and review ontology mapping and its importance in combining distributed and heterogeneous ontologies. Developing ontology mapping facilitates semantic interoperability and improves alignments among the domain ontologies during their design and use. Due to the existence of several ontologies over many domains, ontology mapping serves to achieve semantic correspondence between common elements of different ontologies and interoperability between agents of services [89]. Moreover, matching ontologies increases interoperability between semantic web applications that use different but related ontologies. In this regard, ref. [89] develops a structured-based partitioning algorithm that classifies each ontology's entities into a collection of small clusters and blocks by assigning RDF sentences to those clusters. The authors of [90] aim to automate the discovery of ontology mapping and resolve the instance heterogeneity issue by proposing an approach called risk-minimization-based ontology mapping. Likewise, ref. [109] introduces MAFRA, an ontology mapping framework for distributed ontologies in the semantic web. The suggested framework provides the foundations for managing and executing mapping between distributed ontologies and supporting all parts of the ontology mapping lifecycle. Overall, the body of research aiming at mapping ontologies is extensive; nevertheless, reliance on interactive methods to align ontologies sometimes requires significant human intervention [110]. As a result, it is crucial to automate ontology mapping using novel methods based on artificial intelligence

techniques such as deep learning and decision-tree learning to structurally and semantically ensure valid mappings between similar concepts in two or more distinct ontologies [111]. Another future research direction is improving the existing ontology systems in terms of efficiency and availability and developing systems for large-scale mapping tasks in a linked open data environment [112]. Researchers may also examine the impact of ontological changes, adjustments, and corrections on information systems. This is important because no agency has a perfect ontology to comply with. Therefore, ontology mapping should be an ongoing service and not a one-off process [113]. More practical and innovative solutions for the readjustment of ontologies and data remarking are welcome to minimize the negative impacts of ontology changes on information systems.

### 5.5. Building Information Modeling

The fifth cluster includes studies that focus on the interplay between the semantic web and building modeling information (BIM). According to [114], semantic web technologies are a fruitful addition to existing technologies such as BIM software environments and the IFC (industry foundation classes) specification in EXPRESS, which is a language for defining data structures. The authors of [115] argue that deploying the description language IFC enables BIM systems to provide building information in a widely interoperable format. Due to the complexity and diversity of domain knowledge across BIM and GIS (geographic information systems), syntactic approaches cannot completely exchange semantic information specific to each system. However, through a common language for sharing, as is provided by IFC, it is possible to achieve semantic interoperability between diverse BIM and GIS applications [115]. The authors of [116] note that the lack of interoperability across the BIM and geospatial domains can be resolved by the ability of semantic web technology to convey meaning that is interpretable by construction project participants. Investigating these capabilities of the semantic web, ref. [116] translates a building's elements and GIS data into a semantic web data format and employs a collection of standardized ontologies for construction operations to integrate and query the heterogeneous temporal and spatial data. Similarly, ref. [117] aims to extend the semantic interoperability between BIM and GIS by proposing an approach consisting of three main steps: ontology construction, semantic integration through interoperable data formats and standards, and the query of heterogeneous information sources. The suggested approach helps to enhance data sharing and integration between BIM and GIS. It ensures the seamless integration of building- and construction-related data via a new ontology based on the EXPRESS schema. More recently, ref. [118] reviewed the integration of BIM and IoT devices and emphasized that this combination can realize the maximum potential of using semantic web technologies for the real-time monitoring and assessment of building performance. Even though the extant literature has highlighted the potential of integrating BIM and semantic web technologies to meet the needs for storing, exchanging, and utilizing heterogeneous datasets, a particular focus on how to address information overload in IoT deployments and standardize construction process data, prefabrication data from suppliers, and tracking data from the IoT to comply with local codes and regulations in the construction industry is required [118]. Furthermore, more research is needed to understand the role of semantic web technologies in serving as a critical foundation for building digital twins for the sustainability assessment of constructions [119]. It is vital to look into more practical use cases for the research into the integrated use of BIM and the semantic web in building lifecycles [39]. Information integration and management require standards to minimize the cost and time of sorting large and heterogeneous datasets and ensure the effective development of IoT- and BIM-enabled smart environments [118].

### 5.6. Bioinformatics

The theme in the sixth cluster was labeled "bioinformatics" and consists of 16 articles. With the exponential increase in both the volume and diversity of so-called "omics" data (i.e., proteomics, genomics, transcriptomics), there is a need to develop and adopt data

standards to achieve the promise of systems biology [120]. The interrelated and complex nature of biological data raises several data and tool integration barriers. For this reason, several scholars within the bioinformatics community have studied several interoperability standards, solutions, and projects over the past decades. For example, ref. [106] illustrates the BioMoby project, which aims to standardize methodologies to synchronize information exchange and access to analytical resources through a consensus-driven approach. Unlike other semantic web service interoperability initiatives, this project is realized using a new type of XML schema derived from an ontology, aiming to define the biological intent and syntax of the data passed into and out of a service. The authors of [121] present Semantic Automated Discovery and Integration (SADI), a lightweight approach to simplify the discovery and consumption of semantic web services in bioinformatics and other scientific domains based on foundational web standards. The suggested approach enables bioinformatics software development with new interoperable and integrative behaviors. It helps accurately model the services and the end-user needs for the automated discovery of key services, the integration of the resulting data, and the automation of service pipelines. Similarly, ref. [122] describes another project, Open PHACTS, which aims to deliver and sustain an open pharmacological space based on existing and improved state-of-the-art semantic web standards and technologies. The project also improves drug discovery in academia and industry and drives open innovation and in-house non-public discovery research. To ensure user-oriented semantic service discovery, [123] proposes a data model and lightweight semantic discovery architecture that enables interoperability and composition across different autonomous third-party services, thereby providing a good fit for user requirements in bioinformatics and other domains. Finally, ref. [124] discusses three projects employing semantic web technologies to facilitate services' automated discovery and compositions and enable seamless and transparent interoperability, including Grid, MOBY-Services, and Semantic-MOBY. Overall, the literature highlights that the semantic web promises to foster integration and interoperability among different bioinformatics resources on the web. However, this promise is not broadly realizable in practice, as the reasons for failure may be attributed to the basic difference between semantic web technologies and web services. Therefore, researchers should develop more advanced and scalable solutions that address the deficiencies in the reuse of Universal Resource Identifiers (URIs), the lack of accessibility of bioinformatics data, and the need for large-scale data integration for biological phenomena. Methods and tools that provide a semantics-rich representation of data and efficiently exploit highly heterogenous data are required [125]. Moreover, it is necessary to adjust current computational systems to assimilate and integrate the diversification in bioinformatics research and solve existing knowledge-related and interoperability challenges [126].

### 5.7. Education and E-Learning

The seventh cluster revolves around the possibilities of the semantic web in education and e-learning. As a dynamic field, e-learning is currently dominated by a plethora of learning management systems that need to be enhanced in terms of interoperability and data/resource use. For example, ref. [127] states that the application of linked data principles is a promising solution for the interoperability issues facing the field of technology-enhanced learning. While the web-scale integration of educational resources is challenging to maintain, linked data principles can be used to model and expose metadata of educational resources, services, and APIs, thereby achieving metadata interoperability, services discovery, and data mediation [128]. To develop the educational semantic web, ref. [129] suggests a modular semantic-driven and services-based interoperability framework to open up, exchange, and reuse educational systems' content and knowledge components. In the same vein, ref. [130] describes a novel context-aware semantic learning approach to combine content provision, learning process, and learner personality in an over-arching semantic e-learning framework. This framework leverages XML/RDF technologies to solve technical and pedagogical problems such as data integration, heterogeneity, and lack of pedagogy support and framework. Therefore, the development of personalized adaptive

learning needs semantic-based and context-aware systems to support the interoperability of learning objects and learner models [84]. In sum, several research opportunities emerge from this cluster. For instance, researchers could study the process of semantically improving learning resources by creating new ontologies and prototype systems that accommodate them [131]. Collaboration among various educational entities prompts the development of common semantic models that will be used to represent any type of resource or actor involved in educational and learning processes [132]. Ultimately, studies suggesting novel applications for ontology mapping issues are valuable to extend the existing tools for e-learning, enhance semantic e-learning, and improve the educational and learning processes of individuals involved in the knowledge society [133].

*5.8. Semantic Web Languages*

The last cluster is concerned with semantic web languages such as RDF. Only four publications belong to this cluster. In the first study, ref. [134] presents RDF123. This highly flexible open-source tool translates spreadsheet data to RDF and enables users to create their mapping intuitively and obtain much richer spreadsheet semantics. The authors of [135] propose a new approach to create ontologies based on table analysis. Later, ref. [136] described how to automatically infer the intended meaning of tables and represent them in an RDF-linked data format, thereby enhancing search, interoperability, and integration. The techniques implemented on tables are supported by a new Semantic Message-Passing Algorithm, which utilizes linked open data knowledge to enhance available message-passing schemes. Opportunities for future research on semantic web languages exist, and scholars should propose affordable mechanisms for assessing the interoperability of the semantic web and increasing the filtering of extraneous RDF data [137]. In the future, it may be interesting to develop systems that can boost RDF applications in different areas, such as supply chain management [138]. Moreover, additional experiments with real datasets are recommended to reduce the evaluation cost of the spatial component in RFD queries [139].

Based on the results of this section and the clusters from bibliographic coupling (Table 10), important research gaps related to semantic interoperability need further discussions and investigations (see Table 11).

**Table 10.** Top 10 publications in co-citation clusters.

| Cluster 1 | Cluster 2 | Cluster 3 | Cluster 4 | Cluster 5 | Cluster 6 | Cluster 7 | Cluster 8 |
|---|---|---|---|---|---|---|---|
| [98] | [75] | [73] | [88] | [114] | [106] | | |
| [97] | [145] | [79] | [76] | [105] | [160] | | |
| [140] | [71] | [77] | [106] | [115] | [121] | [127] | |
| [86] | [146] | [78] | [90] | [116] | [122] | [128] | [134] |
| [141] | [66] | [74] | [109] | [81] | [123] | [129] | [136] |
| [98] | [147] | [151] | [89] | [157] | [120] | [84] | [135] |
| [142] | [148] | [152] | [154] | [117] | [161] | [130] | [165] |
| [101] | [102] | [153] | [110] | [158] | [162] | [164] | |
| [143] | [149] | [104] | [155] | [159] | [124] | | |
| [144] | [150] | [103] | [156] | [118] | [163] | | |

**Table 11.** Agenda for future research based on co-citation clustering.

| Themes | Future Research Directions | Related Literature |
|---|---|---|
| Internet of Things | • Investigating the opportunities and challenges of vocabulary recommendation tools in IoT-enabled ecosystems<br>• Examining potential solutions for facilitating the standardization efforts for semantic interoperability in IoT environments<br>• Exploring the role of emerging technologies (e.g., Schema.org) to solve the intrinsic complexity of the semantic web<br>• Investigating the collaboration between academics and practitioners for the development of IoT ontologies that enable intelligent interoperability | [166–168] |

**Table 11.** *Cont.*

| Themes | Future Research Directions | Related Literature |
|---|---|---|
| Conceptualization of the semantic web | • Examining the role of the semantic web to support complex issues associated with flexible, automated, and autonomous systems such as Industry 4.0 systems <br> • Investigating the impact of semantic web technologies on software testing <br> • Studying the potentials and challenges of the geospatial semantic web | [169–171] |
| Semantic web services | • Investigating the applicability of new technologies such as blockchain for the semantic web service composition process <br> • Proposing novel frameworks that integrate the use of AI techniques to compose semantic services and provide efficient solutions to user queries | [21,172] |
| Ontology mapping | • Looking into the role of machine learning techniques to aid in the ontology mapping process <br> • Investigating the challenges brought by the generation of ontology mappings and the solutions to address them <br> • Identifying ways to overcome potential ambiguities in background knowledge and relations among concepts | [71,112,113] |
| Building information modeling | • Examining the integration of BIM with real-time data generated from IoT devices to optimize construction and operational efficiencies <br> • Exploring the capabilities of rule-based reasoning in semantic BIMs <br> • Testing use cases with a high-level digital representation of a real building <br> • Evaluating the potential of new technologies to shape BIM and solve the limitations of its application | [114–116,118,173,174] |
| Bioinformatics | • Developing robust methods for embedding ontologies containing richer semantic information in bioinformatics <br> • Developing programs to acquire datasets of necessity, conduct analyses, and design their summarizations and visualizations according to the requirements of bioinformaticians | [106,121,160,175,176] |
| Education and e-learning | • Designing the architecture of semantic-web-enabled recommender systems in education and e-learning <br> • Proposing semantic-based frameworks that automate e-learning processes based on semantic descriptions <br> • Investigating the challenges related to distance learning through the semantic web, such as accuracy, time limitation, cost, information overload, data security, personalization, and copyright issues | [127,129,177–179] |
| Semantic web languages | • Investigating how semantic web languages can satisfy the knowledge representation requirements and the structural coverage of ontologies | [180] |

## 6. Discussion, Limitations, and Future Research

Given the exponential growth of the literature on the semantic web and interoperability and the dearth of comprehensive and systematic analyses of this subject, there is a pressing need to map the current state of this knowledge. Thus, this study was motivated by a desire to visualize and map the structure and scope of research at the intersection of the semantic web and interoperability, thereby tracing the evolution of this field of knowledge and revealing its intellectual structure. We analyzed 3511 publications selected from the Scopus database over two decades using bibliometrics as a proven and objective methodological approach. We obtained objective results by combining several bibliometric techniques. This is in contrast to subjective techniques, as argued by [181]: "an arbitrary selection of evidence is often not fully representative of the state of the existing knowledge, and the selection of some studies over others ultimately leads to what is known in statistic analysis as a sample selection bias".

We present a broad picture of the research on the semantic web and interoperability using concepts from systematic network analysis, thereby overcoming the inherent problems with conventional methodological approaches, which frequently ignore the dynamics and relationships between authors, publications, and the outlets in which research

works are published. Similarly, ref. [182] asserts that this type of relationship is critical for comprehending the intellectual foundation and evolution of knowledge over time in a scientific field.

There is still much to do in obtaining a functional semantic web that will operate across all knowledge. The schematic of the ontology pyramid demonstrates the current state of the art. By examining all studies on the semantic web and interoperability that have been indexed in the Scopus database since 2000, this article conducted the domain's first comprehensive bibliometric analysis, as far as we are aware. The volume of publications enabled the identification of the most prolific scholars, relevant journals, and major trends and thematic hotspots of this interdisciplinary field of research. For example, a closer examination of the co-authorship network revealed that small scholarly networks characterize the typology of current research.

As such, this structure reflects the dominance of a few scholars who are considered the founders and pioneers of semantic web research. Additionally, such authors may play a crucial role in disseminating knowledge and the advancement of the research field. Given the nature of semantic web research, theoretical underpinnings are often supported by the need to demonstrate the applications of theoretical models. It is conjectured that academics are conducting and advancing their research through collaboration with industrial partners and government organizations, rather than other academics at this stage, due to the need to gather data and focus their research on a specific industry problem. As observed in Table 9, the key application areas of focus include building information modeling, bioinformatics, and the Internet of Things. It is possible that once semantic web and interoperability research is mature in some of these key application areas, there could be a greater linkage between different scholarly networks to explore whether their research is applicable in different industry sectors.

The emergence of the semantic web accelerated the evolution of ontologies and advanced interoperability between entities involved in various complex web services. With increased awareness of the semantic web and interoperability, research on these concepts has been expanding to encompass a variety of domains. The journals that publish the majority of semantic-web-related studies are the *International Journal of Semantic Web and Information Systems*, *Semantic Web*, and the *International Journal of Metadata Semantics and Ontologies*. In terms of country-wise distribution, we found that semantic web research is dominated by countries such as the United States, Germany, and the United Kingdom. In terms of institution-wise distribution, the most productive institution was the National University of Ireland Galway (NUIG), followed by Universidad Politécnica de Madrid and the Digital Enterprise Research Institute (DERI). The latter institute is in Galway, Ireland, and is closely affiliated with NUIG.

For those seeking a better understanding of semantic web research, the influential publications identified in this study may be a starting point for comprehending the field's conceptual foundation. Additionally, the use of a keyword co-occurrence network enables the identification of the critical topics and themes discussed in the semantic web and interoperability research communities. The research was primarily focused on three main themes: (1) language, (2) sensors and computing, and (3) graphs and models. The widespread use of ontologies enables semantically enhanced information processing, as well as enhanced interoperability, expressiveness, and computability. The capabilities of semantic web languages can be extended through the use of tools that optimize and adapt them for a variety of applications. With a large number of seamless sensors and data streams, semantic web technologies can be used to improve stream processing, data integration, and reasoning and support the discovery and reusability of event-based services. Our article makes several contributions to the literature on the semantic web and interoperability. First, we identified the primary research focuses of researchers working at the intersection of the semantic web and interoperability, which continues to generate additional discussions and analysis. We identified that the research focus areas revolve around eight major thematic clusters: (1) the Internet of Things, (2) the conceptualization of the

semantic web, (3) semantic web services, (4) ontology mapping, (5) building information modeling, (6) bioinformatics, (7) education and e-learning, and (8) semantic web languages. The co-citation analysis revealed that the most influential papers concentrate on the vision of the Internet of Things achieving seamless interaction between physical objects and information systems through interoperability. Additionally, studies focus on the semantic web's vision, ensuring that machines can discover, combine, and act on web-based information.

It is not surprising that our analysis showed a strong focus on semantic web and interoperability research in the area of the Internet of Things. With a wide array of sensors, and strong communications infrastructures, it is possible to gather numeric, text, and video-based data in large volumes relatively cheaply. This offers businesses and other organizations the possibility of greater insights into areas such as business performance and customer experience. However, this vision is difficult to realize, often because of the semantic and interoperability challenges between different sensors and systems discussed in this paper, which is why researchers have strongly focused on this topic.

Despite the contribution of this study, several limitations exist. In contrast to [25], which used several data sources, we used a single database. As a result, in comparison to other databases such as Web of Science (WoS) and Google Scholar, citations and relatedness between publications will be more conservative. As a result, future studies may include additional databases to ascertain the validity of our findings. Nonetheless, it is worth noting that the vast majority of publications indexed in databases such as WoS and Scopus are dual-indexed in these two databases. Second, we limited our search to publications written in English. As a result, the scope of coverage may be reduced; consequently, future studies could consider publications in other languages, particularly the languages of the most productive countries (Germany and Italy), in order to assess the generalizability of this study's findings across languages. Third, the study's search query may have missed potentially relevant studies. Researchers may wish to extend the search strings to include a broader list of keywords related to the semantic web and interoperability to address this shortcoming. Finally, the bibliographic coupling is somewhat static and retrospective, as it is centered on the cited publications. As a result, additional bibliometric tools such as document co-citation analysis are required to study the topic in a forward-looking manner.

## 7. Conclusions

The scope and depth of semantic web and interoperability research over the last two decades are highlighted in this review paper. As illustrated in Figure 1, while the volume of publications has largely stabilized over the last decade (2011–2020), there are still numerous obstacles to overcome. While 'semantic web', 'ontology', 'interoperability', and 'semantic interoperability' are the top four key terms (Table 6), the frequency of occurrences for 'semantic web' and 'ontology' is nearly three times greater than the number of occurrences for 'interoperability' and 'semantic interoperability'. This may imply that, while both are critical, interoperability-based experiments and research are more challenging to implement due to their inherent complexity and cost. Despite these obstacles, interoperability should remain a primary research focus.

Additionally, while the top four clusters (Table 9) are centered on the underlying structures/technologies, the two key applications identified are BIM and bioinformatics, which have significant natural drivers. Because most human activity occurs within buildings, there are strong incentives to maximize comfort, minimize energy consumption, and reduce maintenance requirements, all of which are facilitated by the IoT and interoperability. Given that bioinformatics is a critical component of drug discovery/development and disease identification, both critical for maintaining and improving human health, it is unsurprising that BIM and bioinformatics have developed into key application areas within semantic web and interoperability research.

Another critical area in which semantic web technologies and interoperability are influential is facilitating data and information exchange within and across product supply chains in food, pharmaceuticals, medical devices, consumer goods, and automotive indus-

tries. The COVID-19 pandemic has posed significant challenges for stakeholders, and a lack of interoperability makes it challenging to determine where products are located and in what condition they are.

The authors present a series of recommendations based on the findings in this paper.

Recommendation 1—Interoperability should remain a primary research focus, and more research should be undertaken to determine how to overcome the barriers to entry in this area.

Recommendation 2—To prevent stagnation in this important area, academia should forge closer links with industry, governments, and standards organizations to develop more effective models through access to real-world data (web3). This is particularly important for global supply ecosystems, given their length, complexity, and the growing need for enhanced transparency and trust.

The Trace Alliance workgroup, founded by OriginTrail and comprising numerous solution providers, developers, researchers, brands, retailers, and government agencies, will consider applying these research findings and investigate the development of proof of concepts and pilot projects in global supply ecosystems to address interoperability, resilience, and transparency and trust concerns. The authors note that a draft working copy of this paper was previous circulated for comments and was titled "The Big picture on Semantic Web and Interoperability. What we know and what we don't".

**Author Contributions:** All authors contributed equally to this project. All authors have read and agreed to the published version of the manuscript.

**Funding:** This publication has emanated from research supported in part by grants from Science Foundation Ireland under Grant Numbers 16/RC/3918, 12/RC/2289_P2 and 16/RC/3835, and by a grant from the European Union's Horizon 2020 research and innovation programme under Grant Number 958371. Further grants supporting this research were provided by Colorado State University and the McGill Center for the Convergence of health and Economics (MCCHE). Author John G. Keogh was partially funded as chairman of Trace Alliance (1 June 2021 to 31 December 2021).

**Institutional Review Board Statement:** Not applicable.

**Informed Consent Statement:** Not applicable.

**Data Availability Statement:** Not applicable.

**Conflicts of Interest:** The authors declare no conflict of interest. Author John G. Keogh acknowledges his former positions as founding chairman of Trace Alliance and GS1 standards adviser to OriginTrail (2018–2019) and as a former executive at GS1 Canada and Global office (2009–2014).

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
