# Peer review of "Charting Past, Present, and Future Research in the Semantic Web and Interoperability"

_futureinternet, doi:10.3390/fi14060161_

Round 1

Reviewer 1 Report

This article conducts the first comprehensive bibliometric analysis in the field of semantic web with the participation of authors from around the world. The literature review methodology is specific and detailed. The text is well structured and well written.   I found the co-citation clustering analysis extremely interesting and I believe that this article will be useful for the reader.   Some minor comments:

formatting : lines 44, 93, 149, 572, 588

lines 142-146 Sections, use the same format typo: line 455 about sensors and computing Figure 5, missing data on figure

Author Response

Dear Reviewer 1, 

Please find our answers in the attachment. 

Kind regards, 

The authors

Reviewer 2 Report

This review manuscript demonstrates how semantic web development can leverage ontologies to optimize data utilization and facilitate interoperability across data domains. The idea is good; however, many issues and major revisions should be addressed according to the following comments:

1) The overall presentation, readability, and discussion analysis are mandatory. Also, the manuscript suffers from a lot of language problems, please correct the language problems, it is weak from the Grammarly and sequences of events, I catch 63 errors by using a personal program!!. The paper must be proofread very carefully by a native speaker or a proofreading agent.

2) In the manuscript title, please try to power the title with strong words. I can't accept the main title with the pronoun "we".   

3) The "Abstract" section should be more intensively focused on the main idea directly and must contain the current problems, their remedy, recommendations, and future vision of this review manuscript.

4) The "Introduction" section should be made much more impressive by highlighting the review contributions. Note that, the introduction section should consist of three parts, i.e., a general introduction to the topic, followed by a comprehensive literature survey for various applications of semantic web technology, then the recommendation and future plan clarifications.

5) The introduction section should be enriched using the up-to-date references 2022, by adding and citing papers in the area of the latest trends of using semantic web technology in autonomous systems and to predict how semantic web technology will be used in Artificial Intelligence, Machine Learning, and IoT. E.g.,

1)Semantic Web and Knowledge Graphs for Industry 4.0 
2) Reliable Deep Learning and IoT-Based Monitoring System for Secure Computer Numerical Control Machines
3) Effective IoT-based Deep Learning Platform for Online Fault Diagnosis of Power Transformers
4) Knowledge-Based Fault Diagnosis in Industrial Internet of Things: A Survey.

6) It is mandatory to add various equations of different machine learning or artificial intelligence techniques with its reference's citation with the help of the abovementioned suggested papers. Also, in subsection 5.1, the authors can add some engineering applications on IoT using the above-suggested references.

7) The resolution and quality of all figures must be modified; they should be presented as close to the camera-ready format.

8) Please the editor and authors should check carefully all figures if there is any photo or table taken from another published paper. It should be taken permission from the other publisher association. It is a very important issue.

9) It is mandatory to check carefully all the abbreviation definitions, symbols, and standard units in the whole manuscript. (e.g., IoT, GIS, URLs, and so on).

10) It will be helpful to the readers if some discussions about the insight of the main survey are added as remarks for more explanation and comparisons.

11) The conclusion section should be rearranged, and the authors' recommendations should be pointed out. Also, the authors may propose some interesting problems as future work at the end of the conclusion.

Author Response

Dear Reviewer 2, 

Please find our answers in the attachment. 

Best regards, 

The authors

Round 2

Reviewer 2 Report

Most of my comments are adjusted. A lot of self-citations are found, please the authors must reduce them.